# Genetic analysis of cis-enhancers associated with bone mineral density and periodontitis in the gene *SOST*

**Avneesh Chopra**[1][☉], **Jiahui Song**[1][☉], **January Weiner 3rd**[2], **Dieter Beule**[2], **Arne S. Schaefer**[1]*

**1** Department of Periodontology, Oral Medicine and Oral Surgery, Institute for Dental and Craniofacial Sciences, Charité-University Medicine Berlin, corporate member of Freie Universität Berlin, Humboldt-Universität zu Berlin, and Berlin Institute of Health, Berlin, Germany, **2** Core Unit Bioinformatics, Berlin Institute of Health, Berlin, Germany

☉ These authors contributed equally to this work.
* arne.schaefer@charite.de

## Abstract

A haplotype block at the sclerostin (*SOST*) gene correlates with bone mineral density (BMD) and increased periodontitis risk in smokers. Investigating the putative causal variants within this block, our study aimed to elucidate the impact of linked enhancer elements on gene expression and to evaluate their role in transcription factor (TF) binding. Using CRISPR/dCas9 activation (CRISPRa) screening in SaOS-2 cells, we quantified disease-related enhancer activities regulating *SOST* expression. Additionally, in SaOS-2 cells, we investigated the influence of the candidate TFs CCAAT/enhancer-binding protein beta (CEBPB) on gene expression by antisense (GapmeR) knockdown, followed by RNA sequencing. The periodontitis-linked SNP rs9783823 displayed a significant cis-activating effect (25-fold change in *SOST* expression), with the C-allele containing a CEBPB binding motif (position weight matrix (PWM) = 0.98, $P_{corrected}$ = 7.7 x $10^{-7}$). *CEBPB* knockdown induced genome-wide upregulation but decreased epithelial-mesenchymal transition genes (P = 0.71, AUC = 2.2 x $10^{-11}$). This study identifies a robust *SOST* cis-activating element linked to BMD and periodontitis, carrying CEBPB binding sites, and highlights *CEBPB*'s impact on epithelial-mesenchymal transition.

## Introduction

Bone remodeling is a process that adjusts the distribution of bone mass to meet multiple demands throughout life, including replacing damaged or old bone with new bone. To adapt bone mass to specific demands, cells integrate mechanical and metabolic environmental cues to balance bone formation and resorption [1]. Specifically, the loading of bone by dynamic mechanical forces is "sensed" by mature osteocytes within the mineralized matrix. Mechanical loading results in the suppression of the *SOST* gene, an inhibitor of the Wnt/β-catenin pathway specific to bone tissue, whereas bone unloading induces osteocytes to synthesize and secrete SOST, which promotes osteoclast genesis and results in reduced bone formation and bone mass loss [2]. Thus, SOST is a negative regulator of bone formation. Similarly,

**Data availability statement:** All relevant data are within the paper and its Supporting Information files.

**Funding:** The author(s) received no specific funding for this work.

**Competing interests:** The authors have declared that no competing interests exist.

numerous genome-wide association studies (GWAS) have shown genetic associations of *SOST* with BMD and fractures [3–9], and increased serum levels of SOST correlate with decreased BMD [10, 11]. In addition, GWAS have also reported genetic associations of *SOST* with high-density lipoprotein cholesterol and triglyceride levels [12–16]. Similarly, higher serum SOST levels correlate positively with serum triglyceride levels [17]. In addition, several GWAS have also found an association between *SOST* and blood eosinophil counts [18–20], white blood cells whose functions include modulation of inflammatory responses as well as anti-parasitic and bactericidal activity. Similarly, a role for SOST in the control of inflammation has been reported in mouse models where SOST inhibited tumor necrosis factor alpha (TNFA)-induced innate immune responses [21,22]. These findings suggest a more complex role for SOST with a function in bone metabolism, but also in energy and inflammatory signaling (reviewed in [23].

*SOST* transcript and protein levels were also found to be significantly elevated in gingival tissue from patients with the oral inflammatory disease periodontitis [24], a complex common disease diagnosed by progressive alveolar bone loss as a result of oral inflammation [25]. Periodontitis has been epidemiologically associated with BMD in adolescents [26], young adults [27], and the elderly [28]. In a recent study, we found a common haplotype block in the genetic region of *SOST*, which was associated with BMD (rs1513670) [6] and severe periodontitis (classified as periodontitis stages III/IV, grade C; PIII-IV/C) diagnosed in young adults (rs6416905, linkage disequilibrium $r^2 > 0.8$) [29]. Of note, the genetic association with periodontitis was found in a gene x smoking interaction study and only in tobacco smokers, resulting in an increased PIII-IV/C risk in smokers but not in nonsmokers. The common association of this haplotype block implies pleiotropic effects on BMD and periodontitis in smokers. Notably, tobacco smoking renders bone susceptible to osteoporosis by causing an imbalance in the mechanisms of bone turnover, resulting in lower bone mass and BMD, as recently reviewed [30].

Knowledge of the molecular genetic mechanisms underlying this common association would help to better understand the common etiology of the inflammatory and metabolic bone diseases periodontitis and osteoporosis. Therefore, this study aimed to find the putative causative variant(s) of this association and to identify the molecular mechanism contributing to the increased disease risk and mediating the putative impaired *SOST* regulation.

## Materials and methods

### Screening for functional variants associated with periodontitis

LD ($r^2 > 0.8$) between the BMD GWAS lead SNP rs1513670, the periodontitis leads SNP rs6416905, and other common SNPs in this haplotype block was assessed using LDproxy [31] with genotypes from the North-Western European populations CEU [Utah Residents with Northern and Western European Ancestry] and GBR [British in England and Scotland] [32] from the International Genome Sample Resource (IGSR) (**Appendix** Table 1). We analyzed whether these SNPs were mapped to chromatin elements that correlate with regulatory functions of gene expression provided by ENCODE [33] (Fig 1). These elements were 1) open chromatin as determined by DNAse I hypersensitivity (DHS), 2) histone modifications H3K27Ac and H3K4Me1, 3) transcription factor binding sites (TFBS) experimentally confirmed by chromatin immunoprecipitation sequencing (ChIP-seq), and 4) chromatin state segmentation. We used the Transcription Factor Affinity Prediction (TRAP) Web Tools to bioinformatically determine whether the SNP sequences of the associated variants contained conserved regulatory binding sites [34]. We used the TRAP multiple sequence module with the vertebrate TF matrices from TRANSFAC and Jaspar and the 'human promoters'

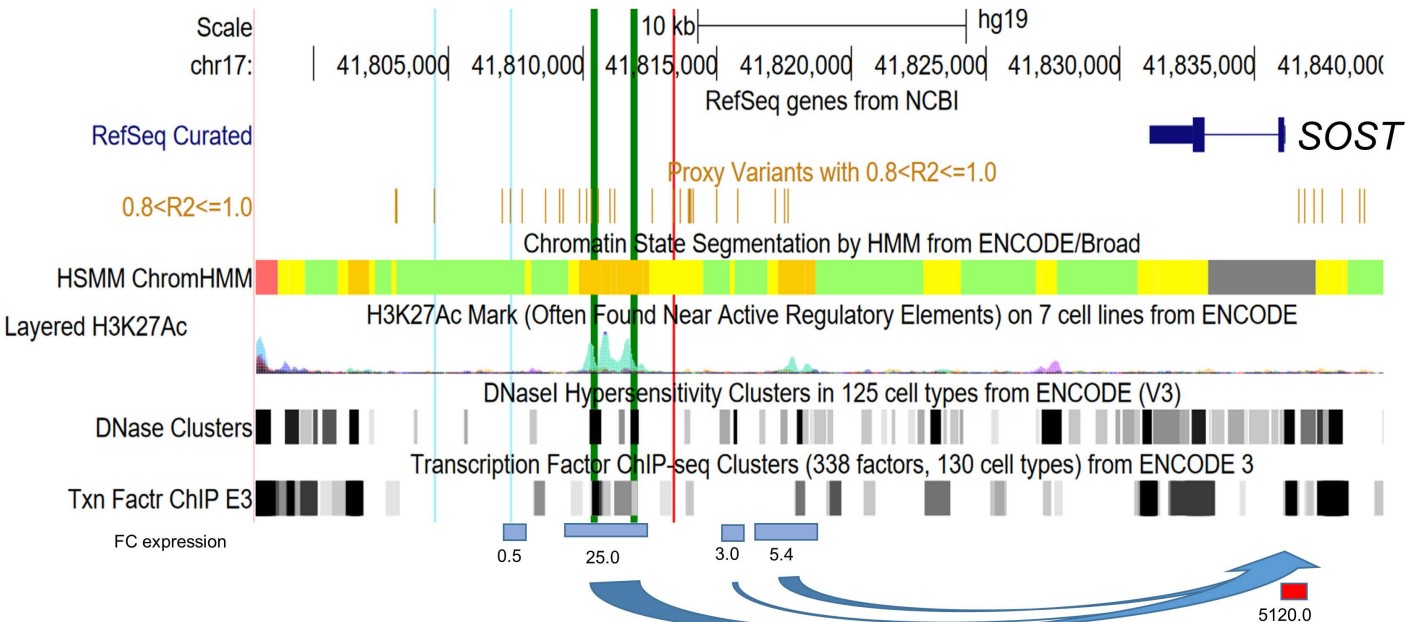

**Fig 1. Overview of the genetic region associated with bone mineral density (BMD) and periodontitis at the gene *SOST*.** Cis-regulatory effects on *SOST* expression containing functional CEBPB binding sites (the order of proxy SNPs from left to right is shown in Appendix Table 1). Light blue vertical lines mark the GWAS lead SNPs rs6416905 (periodontitis, left) and rs1513670 (BMD, right). Bold green vertical lines mark the CEBPB binding sites confirmed by ChIP-Seq experiments in lung fibroblasts of Caucasian origin. A red vertical line marks the putative functional SNP rs9783823. The bottom panel shows the fold change (FC) of *SOST* expression induced by CRISPRa in the current study using sgRNAs at the positions indicated by the horizontal bars (blue bar = binding in enhancer regions, red bar = binding in promoter region). The information in the other panels was obtained from the UCSC Genome Browser (hg19) using ENCODE 3 data (color code for enhancers in the ChromHMM panel: orange = strong enhancer, yellow = weak enhancer).

background model [35] with the reference and alternative allele SNP sequences as input and ranked the output by P value. In the exploratory screen, we set a multiple-test corrected P value threshold of $P < 5 \times 10^{-6}$ to reduce the number of false-positives. P values were corrected for multiple tests using the Benjamini-Hochberg method. We quantified the effects of the reference and alternative SNP alleles on the binding affinities of the TFs using the sTRAP software tool [36]. The predicted TFBS that showed allele-specific differences in binding affinities were then manually checked for alignment of the SNP sequence with the specific TF motif, as indicated by the PWM. The matrix profiles for the predicted TFs were obtained from JASPAR (version 2022) [37]. If the TF matrix profile was congruent with the DNA sequence at the SNP, a TF that showed significant binding sites at the SNP sequences as well as significant differences in binding affinities between the SNP alleles was considered indicative of a TF worthy of further molecular biological investigation.

## Cell culture and transfection

For CRISPRa experiments, HeLa cells were cultured as recently described [38]. HeLa cells were used because CRISPRa transfection can be efficiently performed in these cells and they have a high post-transfection survival rate. Briefly, cells were cultured in Earle's MEM medium containing 10% fetal bovine serum, 2 mM L-glutamine, 1% non-essential amino acids, and 1% penicillin-streptomycin. HeLa cells were seeded at 80,000 cells per 6-well (Techno Plastic Products, TPP) to achieve 50-60% confluence during transfection using jetPEI transfection reagent (Polyplus-transfection). For reporter gene assays, *CEBPB* knockdown, and overexpression experiments, we used the human bone osteosarcoma cell line SaOS-2, because in

this osteoblast-like cell line, *SOST* is more highly expressed than in HeLa, suggesting that the transcriptional machinery, including TFs, that regulate *SOST* expression is present in this cell line. SaOS-2 cells were cultured in complete growth medium Alpha MEM Eagle (PAN-Biotech GmbH) supplemented with 10% fetal bovine serum and 50 mg/mL gentamicin. Prior to transfection with Lipofectamine 2000 (Thermo Fisher Scientific), SaOS-2 cells were seeded at 250,000 cells per well in a 6-well plate format for the reporter gene assays and *CEBPB* overexpression experiments. For the GapmeR knockdown experiment, SaOS-2 cells were seeded at a density of 100 cells per 6-well with Lipofectamine 2000 one day prior to GapmeRs (single-stranded antisense oligos) transfection. All cell lines used in this study were purchased from ATCC and maintained at the Dental Clinic of Charité - Universitätsmedizin Berlin.

## Electrophoretic mobility shift assay (EMSA)

To determine allele-specific TF binding to the SNP DNA sequence, we performed EMSAs using the Gelshift Chemiluminescent EMSA Kit (Active Motif) as recently described [38]. Briefly, double-stranded oligonucleotides for each SNP allele, flanked by 21 bp up- and downstream in 3′-biotinylated and unbiotinylated forms, were obtained by annealing with their respective complementary primers (**Appendix** Table 2). Nuclear protein extract was prepared from SaOS-2 cells using the NE-PER Nuclear and Cytoplasmic Extraction Kit (Pierce Biotechnology). For supershift EMSA, 20 fmol of biotin-labeled oligonucleotide probes were incubated with 10 µg of nuclear protein extract in 1x binding buffer and 2 µL (10 µg/50 µL) of a specific monoclonal antibody (C/EBP beta (H-7): sc-7962; GFI-1 (B-9): sc-376949, both from Santa Cruz Biotechnology, Inc.) for 20 min at room temperature. To verify the specific DNA-protein interaction, 4 pmol of unlabeled oligonucleotides were added to the binding reaction. The DNA-protein complexes were electrophoresed in a 5% native polyacryl-amide gel in 0.5x TBE buffer at 100 V for 1 h and visualized by chemiluminescence detection (Chemostar Touch, INTAS) after electrotransfer and crosslinking of the products on a nylon membrane. Band intensities were quantified by the absolute value area of the shifted antibody bands using the ImageJ software [39]. At least 2 technical replicates of each EMSA were tested (**Appendix**).

## Luciferase-based reporter gene assays

The DNA sequences spanning the putative causal SNPs were cloned into the firefly lucif-erase vector pGL4.24 (Promega) to subsequently test their regulatory potential on reporter gene expression. The reporter gene plasmids containing either the reference or alternative allele within the TF binding motif differed by a single nucleotide and were amplified by PCR. Primer sequences are listed in **Appendix** Table 3. The PCR products were inserted into the *Hin*dIII restriction site of the reporter plasmids upstream of the minimal promoter. Reporter plasmids were amplified in 5-alpha competent *E. coli* (NEB) and extracted using the QiaPrep Plasmid Mini Kit (Qiagen).

SaOS-2 cells were co-transfected in triplicates with 2.7 µg firefly luciferase reporter plasmid carrying the allele-specific SNP sequence together with 0.3 µg *renilla* luciferase reporter vector phRL-SV40 (Promega) in 6-well plates. As controls, SaOS-2 cells were transfected with the empty pGL4.24 and phRL-SV40 plasmids. After 24 hours, firefly and *renilla* luciferase activi-ties were quantified using the Dual-Luciferase Stop & Glo Reporter Assay System (Promega) with a luminometer (Orion II Microplate, Berthold). Reporter plasmid activities were quan-tified as relative light units and normalized as the ratio of luciferase to *renilla* activity to cal-culate relative fold changes. Differences in transcript levels were calculated using a two-sided T-Test with GraphPad Prism 6 software (GraphPad Software, Inc.).

## CRISPR/dCas9 activation (CRISPRa) & quantitative real-time PCR (qRT-PCR)

Assigning the regulatory effects of associated regions on specific genes based on public eQTL data and linear proximity to the nearest gene is prone to error, as eQTLs are statistical observations without direct molecular evidence [40], and enhancers are often mapped at great distances from their actual targets and do not necessarily affect the expression of the nearest gene. Induction of *SOST* expression by positioning an activator protein at the sites of the associated SNPs using the CRISPRa system allows quantification of the effects of putative enhancers on *SOST* expression and provides direct evidence of cis-regulatory function. CRISPRa allows specific and efficient quantification of the regulatory potential of a genomic sequence on the expression of candidate target genes at physiological concentrations and in the endogenous chromosomal context including naturally occurring genetic variants [41]. Therefore, we used CRISPRa to analyze whether the associated regions have cis-regulatory effects on *SOST* expression. To test cis-regulatory effects, we aligned the 36 linked haplotype SNPs to the chromosomal locations and integrated chromatin elements from ENCODE (ENCODE-Project-Consortium 2012) that correlate with regulatory functions of gene expression characterized by DNAse I hypersensitivity, H3K4Me1 methylation, and TF ChIP-Sequencing clusters determined by the ENCODE 3 project [42] (Fig 1). We designed 9 sgRNAs to target the putative regulatory elements in the intergenic region downstream of *SOST*, which carried the linked SNPs. 4 sgRNAs were designed to target the promoter region of *SOST*, which also carried linked SNPs. SgRNAs were designed using the E-CRISP online tool [43]. A scrambled sgRNA with no genomic target was used as a negative control, as described in [38]. The *SOST* sgRNAs and chromosomal locations are listed in the **Appendix** Table 4. For CRISPRa of *GFI-1*, we used the best-functioning sgRNA (fw (5'-3'): CACCGCAAAAT-TAAAGGCCGCGCGG; rev (5'-3'): AAACCCGCGCGGCCTTTAATTTTGC) targeting the *GFI-1* promoter upstream of -70 transcription start site (TSS).

We cloned the sgRNA sequences into the sgRNA(MS2) vector (plasmid #61424) at the *Bbs*I site according to the protocol described in [44]. The CRISPRa system was transfected into HeLa cells. For CRISPRa, each 6-well was transfected with 1 μg sgRNAs(MS2) (containing specific sgRNA(s)), 1 μg dCAS9-VP64_GFP [plasmid #61422], and 1 μg MS2-P65-HSF1_GFP [plasmid #61423] and incubated for 44 h. All plasmids were obtained from Addgene (cloning backbone plasmids was a gift from Feng Zhang) [45]. For quantitative analysis of specific CRISPRa, we used qRT-PCR. Total RNA was extracted from human cells using the RNeasy Mini Kit (Qiagen) according to the manufacturer's instructions. Complementary DNA (cDNA) was synthesized from 500-1,000 ng DNA-free total RNA, using the High-Capacity cDNA Reverse Transcription Kit (Applied Biosystems). SYBR Select Master Mix (Applied Biosystems) was used with the following primers: *GAPDH:* 5'-CAAATTCCATGGCAC CGTCA-3', 5'-CCTGCAAATGAGCCCCAG-3', *SOST:* 5'-GCTGGAGAACAACAAGACCA-3', 5'-GTAGCGGGTGAAGTGCAG-3', *CEBPB:* 5'-AGCGACGAGTACAAGATCCG-3', 5'-AGCT GCTCCACCTTCTTCTG-3', *GFI-1:* 5'-CCGCGCTCATTTCTCGTCA-3', 5'-ACGGAGG GAATAGTCTGGTCC-3'. Relative fold changes in gene expression were calculated by the $2^{-\Delta\Delta Ct}$ method with *GAPDH* as the reference gene. Statistical differences in transcript levels were calculated using a T-Test.

## Sequencing of rs9783823 in IMR90 and A549 cells

DNA was extracted (Blood & Cell Culture DNA Mini Kit, Qiagen) from cell pellets of IMR90 and A549 cells provided by Prof. Clemens Schmidt, Cancer Genetics and Cellular Stress Responses Group, Max Delbrück Center for Molecular Medicine (MDC), Berlin, Germany. A

region up and downstream of rs9783823 was amplified by PCR (5'-CGAGTCT CACTTCCTACCTCA-3', 5'-AGAATCACCTGGAGAGCTGT-3'), and the 279 bp PCR product was sequenced in both directions by the Sanger method at LGC Genomics, Berlin.

## CEBPB knockdown in SaOS-2 cells and RNA sequencing

LNA GapmeRs either targeting unique regions of *CEBPB* isoforms or not targeting any region (scrambled, used as negative control) were designed by Qiagen. A mixture of 3 LNA GapmeRs or one scrambled LNA GapmeR was used. The GapmeRs hybridized to unique regions of *CEBPB* isoforms as follows: Gene Globe ID LG00828145-DDA, LG00828146-DDA, LG00828147-DDA. LNA GapmeRs were transfected into SaOS-2 cells at a final concentration of 225 µM for 48h in biological triplicate. Total RNA was extracted using the RNeasy Mini Kit (Qiagen, Germany). 500 ng total RNA from transfected cell cultures were sequenced on a NextSeq 500 with 16 million reads (75 bp single end) using the NextSeq 500/550 High Output Kit v2.5 (75 cycles). RNA sequencing was performed at the Berlin Institute of Health, Core Facility Genomics. Reads were aligned to the human genome sequences (build GRCh38.p7) using the STAR aligner v. 2.7.8a [46]. Quality control (QC) of the reads was inspected using the multiqc reporting tool [47], which combines several approaches, including fastqc (available online at http://www.bioinformatics.babraham. ac.uk/projects/fastqc), dupradar [48], qualimap [49], and RNA-SeqC [50]. Raw counts were extracted using the STAR program. Differential gene expression was performed using the R package DESeq2 [51], and version 1.30 was used. Gene set enrichment was performed with the CERNO test from the tmod package [52], version 0.50.07, using the gene expression profiling-based gene set included in the package, and MSigDB [53], v.7.4.1. The goseq package, version 1.38 [54], was used for the hypergeometric test and the Gene Ontology gene sets. Reads, raw counts, and the results of the differential expression analysis have been submitted to the Short Read Archive via the Gene Expression Omnibus (GEO accession number GSE269019). The P values of differentially expressed genes were corrected for multiple tests using the Benjamini-Hochberg correction. The corrected P values are given as q values (false discovery rate [FDR]).

## *CEBPB* overexpression in SaOS-2 cells

*CEBPB* isoforms LAP2 and LIP were overexpressed in SaOS-2 cells using episomal expression plasmids (pCMV-FLAG LAP2 (#15738); pCMV-FLAG LIP (#15737)) to determine whether *CEBPB* affects *SOST* expression. These plasmids were obtained from Addgene (gifted from Joan Massague). The empty pGL4.24 vector was used as a control plasmid to normalize the overexpression experiment. 3 µg of each vector was transfected into SaOS-2 cells and incubated for 48h, followed by RNA extraction and qRT-PCR.

## Results

### CRISPRa screen identifies strong cis-regulatory elements downstream of *SOST* associated with alveolar bone loss

SNPs rs1513670 and rs6416905 were lead SNPs showing an association with BMD and periodontitis in smokers [6,29]. These two SNPs were in strong LD ($r^2 > 0.8$) with each other and with 34 additional SNPs (**Appendix** Table 1), spanning the transcribed region of *SOST*, including a large intergenic region downstream of the 3'UTR (Fig 1). We first identified disease-associated regulatory cis-regulatory chromatin elements that are in strong LD with the associated SNPs. To this end, we transfected HeLa cells with the CRISPRa system, which encodes sgRNAs designed to direct the SAM activation system to chromatin elements that

exhibit biochemical enhancer marks such as DHS, H3K27Ac, and H3K4Me1 histone modifications, as well as TFBS experimentally confirmed by ChIP-Seq (provided by ENCODE; Fig 1). We chose HeLa cells because this cell line allows efficient CRISPRa plasmid transfection and does not show *SOST* expression according to the Human Protein Atlas (https://www.proteinatlas.org/ENSG00000167941-SOST/cell±line).

The strongest induction of *SOST* expression was observed with sgRNAs targeting the *SOST* promoter 270 bp and 1,100 bp upstream of the TSS, with a 5,119-fold and 56-fold increase in *SOST* expression, respectively. This demonstrated the functionality of our CRISPRa system to induce *SOST* expression. An associated enhancer located 12.9 kb downstream of the *SOST* 3'UTR showed a 25-fold increase in *SOST* transcript levels. The second strongest associated enhancer showed a 5-fold increase in expression and was located 11.6 kb downstream of the *SOST* 3'UTR. H3K27Ac marks were most prominent in these regions (ENCODE) (Fig 1). The other predicted enhancers marked by associated SNPs showed a < 3-fold increase in *SOST* expression. These regions showed much weaker biochemical marks of active regulatory elements or no enhancer marks.

## Bioinformatic analysis of associated SNPs predicts allele-specific effects on TF CEBPB and GFI-1 binding affinities in functional cis-regulatory elements

14 associated SNPs in strong LD ($r^2 > 0.8$) were located on the enhancer shown in our CRISPRa screen to have the strongest cis-regulatory effect on *SOST* expression. To identify biologically functional SNPs, we analyzed whether the different alleles of these 14 SNPs have any predicted effects on the binding affinities of TFs (**Appendix** Table 1). The highest-ranked TFBS was found for rs9783823 (**Appendix** Table 5). This SNP was predicted to bind CEBPB ($P_{combined} = 7.7 \times 10^{-7}$, PWM = 0.98). The CEBPB motif was predicted to have strong binding affinity on the reference C-allele ($P < 3.29 \times 10^{-6}$), but not on the alternative T-allele ($P = 0.013$). The ancestral (reference) rs9783823-C allele of the CEBPB motif was less common in European populations (C = 0.398 EUR) compared to African populations (C = 0.608 AFR). TF ChIP-Seq clusters (338 factors, 130 cell types) from ENCODE 3 also showed binding of CEBPB in the native human chromatin in this region. ENCODE reported 2 CEBPB ChIP-Seq peaks at chr17:41810335-41810590 (hg19) (peak 1) and chr17:41811788-41812043 (peak 2). These data were taken from ENCODE, which was generated in cultured white Caucasian lung fibroblasts (IMR90 and A549 cells) (**Appendix Material**). The CEBPB binding motifs at peak 1 and peak 2 were predicted with $P < 0.00041$ and $P < 0.00083$ (CEBPB motif +/- 5 bp), respectively. The C-allele of the CEBPB binding sites at peak 1 and peak 2 are located at chr17:41810515 and chr17:41811942 (hg19), respectively. The rs9783823-C allele at the CEBPB binding site is located at chr17:41813369. Notably, this resulted in a distance of exactly 1,427 bp between the C-alleles of the CEBPB motifs of ChIP-Seq peaks 1 and 2, and also 1,427 bp between the C-alleles of the CEBPB motifs of ChIP-Seq peak 2 and rs9783823. We next investigated the possible reason for the absence of a CEBPB-specific ChIP-Seq peak at rs9783823. The lung fibroblast cell lines IMR90 and A549 were derived from white Caucasians. In this genetic ancestry, the reference allele rs9783823-C of the CEBPB motif is the minor allele (C|C = 0.155, C|T = 0.485, and T|T = 0.360; EUR population, ENSEMBL). We sequenced rs9783823 in IMR90 and A549 cells and found that both cell lines were homozygous for rs9783823-T (**Appendix** Fig 1), indicating a loss of the CEBPB motif. In addition, our TFBS analysis revealed a conserved regulatory binding site of the transcriptional repressor GFI-1 for SNP rs8071941 ($P = 0.002$) and a predicted difference in binding affinity between the two alleles of $\log(P) = -3.14$. The TFBS prediction was not significant after correction for multiple testing and did not pass the

pre-assigned significance threshold of $p < 5 \times 10E-6$. However, the DNA sequence at rs8071941 matched the GFI consensus binding motif, and the GFI-1 binding site of SNP rs8071941 was located on CEBPB ChIP-Seq peak 1, 195 bp upstream of the C-allele of the CEBPB motif. Subsequently, we also searched the DNA sequence of peak 2 for GFI-1 binding sites and observed a second GFI-1 binding site on ChIP-Seq peak 2, 81 bp upstream of the C-allele of CEBPB. GFI-1 binding at this site was predicted by TRAP with $P < 0.0014$ (observed GFI-1 motif +/- 5 bp; rank 1). On this enhancer region, we found no other TF motif with significantly different binding affinities between an associated SNP allele and the congruence of the predicted TF motif and the DNA sequence of the SNP. The enhancer (chr17:41816483-41820007, 3.525kb; hg19) that showed the second strongest cis-regulation of *SOST* in our CRISPRa screen had 3 associated SNPs (rs2076793, rs9303540, and rs9908933). These SNPs did not show TFBS with predicted differences in binding affinities between the two SNP alleles with a match of the DNA sequence of the SNPs and the TF motif.

### rs9783823-T impairs TF CEBPB binding

To validate the predicted CEBPB binding at rs9783823, we performed EMSAs with CEBPB antibody and rs9783823 allele-specific DNA probes using nuclear protein extract from SaOS-2 cells. In this cell line, *CEBPB* is moderately expressed (nTPM = 30.4; Human Protein Atlas). The binding of CEBPB antibodies to probes specific for the T-allele of rs9783823 reduced CEBPB binding by 14% to 23% compared to probes with the C-allele (Fig 2, **Appendix** Fig 2).

We then investigated whether the alleles at rs9783823 affect gene activity using luciferase reporter genes in SaOS-2 cells. In the background of the rs9783823 C-allele, reporter luciferase activity was 3.8-fold higher compared to the T-allele ($P < 0.0001$). The rs9783823-C allele increased luciferase activity 2.3-fold compared to the empty plasmid ($P < 0.0001$), whereas the alternative T-allele showed no increase in luciferase activity (-1.5-fold; Fig 3). Consistent with our experiments, the GTEx data also indicated that rs9783823 is a biologically functional SNP that correlated with regulatory effects on *SOST* expression. GTEx observed that rs9783823 had the strongest eQTL effect on *SOST* expression compared to other genes, with reduced *SOST* expression in the genetic background of the T-allele (normalized effect size $\beta = -0.17$, $P = 5.1 \times 10^{-13}$; observed in the artery.

### *CEBPB* knockdown represses genes involved in epithelial-mesenchymal transition and extracellular matrix (ECM) interaction in SaOS-2 cells

*CEBPB* mutant mouse strains showed delayed bone formation with suppression of osteoblast differentiation [55]. *CEBPB* loss-of-function mice showed increased bone resorption and dysregulated expression of *CEBPB* affects bone mass [56]. Therefore, we were interested in investigating the effects of *CEBPB* knockdown in SaOS-2 cells. Silencing *CEBPB* expression in SaOS-2 cells using a mixture of 3 LNA GapmeRs resulted in a 64% reduction of *CEBPB* expression (determined by qRT-PCR, Fig 4A). Genome-wide expression profiling by RNA-Seq after *CEBPB* knockdown using a mixture of 3 LNA GapmeRs revealed 3,787 upregulated genes with $\log_2$ fold change ($\log_2$FC) $\geq 2$) and 604 downregulated genes with $\log_2$FC $< -1$ ($P_{adj} < 0.05$; Fig 4B). The most upregulated protein-coding gene with the lowest P value was the gene *nitric oxide synthase 3* (*NOS3*) ($\log_2$FC = 11.31, $P = 6 \times 10^{-21}$) and the most downregulated protein-coding gene was the gene *keratin 17* (*KRT17*) ($\log_2$FC = -6.1, $P = 4 \times 10^{-8}$; Table 1). Using qRT-PCR, we observed a weak increase, however not significant, of *SOST* relative transcript levels (Fig 4C). Our RNA-Seq data showed that *CEBPB* knockdown correlated with a weak increase in *SOST* expression ($\log_2$FC = 0.79) with $P_{adj} = 4.7^{-26}$ (Fig 4D).

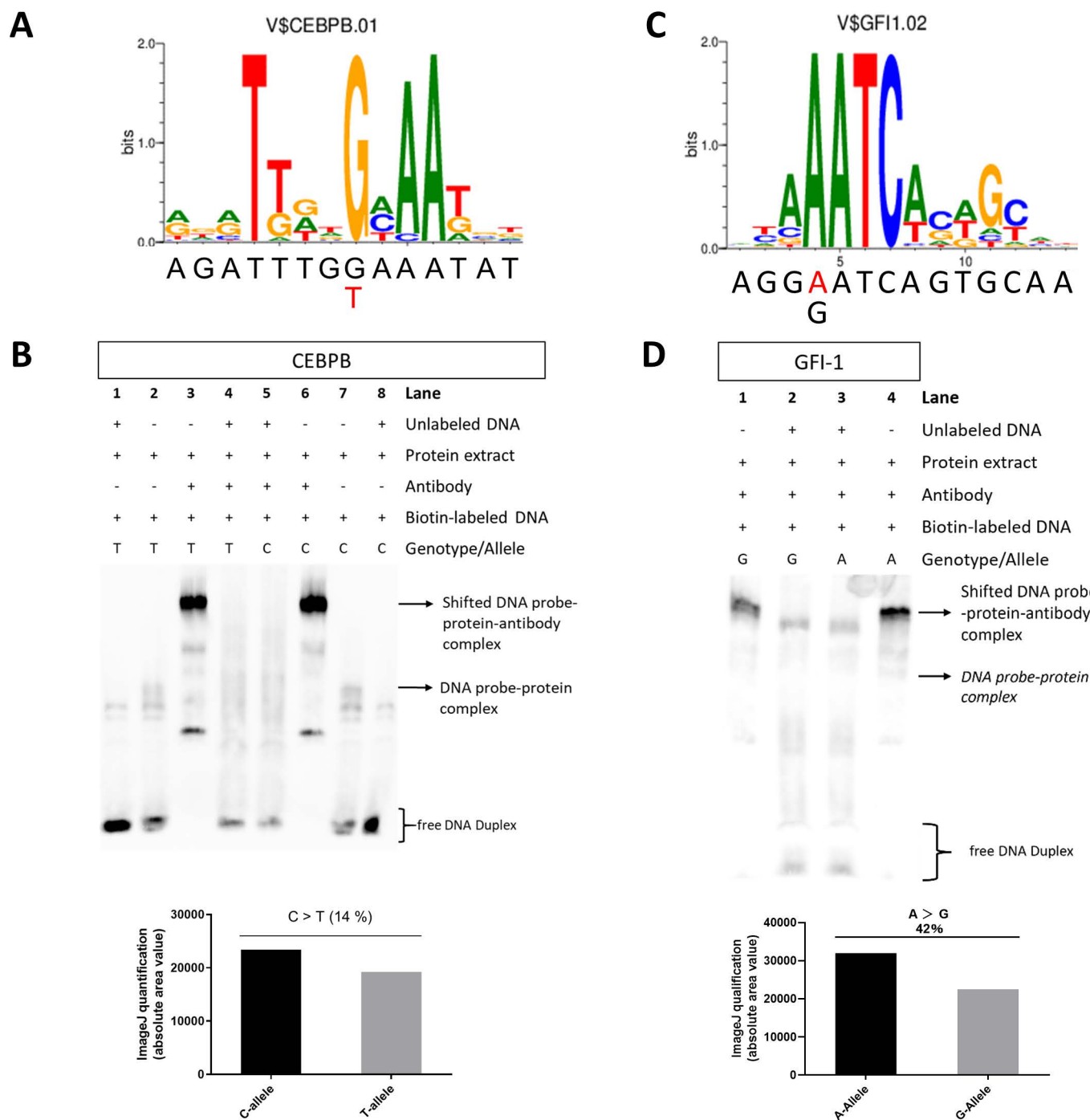

**Fig 2. rs9783823-G and rs8071941-A alleles provide binding motifs for the TFs CEBPB and GFI-1.** The alternative rs9783823-T allele reduces the predicted affinity of CEBPB binding (**A**). CEBPB binding at the SNP sequence was detected by antibody EMSA, which showed a 14% (**B**) to 23% (**Appendix Fig 2**) reduction in CEBPB binding in the presence of the alternative T-allele. The rs8071941-G reference allele reduces the predicted affinity of GFI-1 binding (**C**), validated by antibody EMSA showing a 42% reduction in GFI-1 binding compared to the alternative A-allele (**D**).

**Table 1. Top 10 up- and down-regulated protein-coding genes after *CEBPB* knockdown in SaOS-2 cells for 48 h.**

|  | Gene | Entrez | Description | Log2FC | lfcSE | adj.P |
|---|---|---|---|---|---|---|
| up | *NOS3* | 4846 | nitric oxide synthase 3 | 11.314 | 1.185 | 5,90E-21 |
|  | *ZNF560* | 147741 | zinc finger protein 560 | 11.284 | 1.187 | 8,30E-21 |
|  | *TUSC8* | 400128 | tumor suppressor candidate 8 | 10,87 | 1.191 | 3,00E-19 |
|  | *F13B* | 2165 | coagulation factor XIII B chain | 9.602 | 1.207 | 6,40E-15 |
|  | *CERS3* | 204219 | ceramide synthase 3 | 9.594 | 1.202 | 5,10E-15 |
|  | *LRRC14B* | 389257 | leucine rich repeat containing 14B | 9.557 | 1.205 | 7,80E-15 |
|  | *ANHX* | 647589 | anomalous homeobox | 9.437 | 0,651 | 1,10E-46 |
|  | *CYP2C9* | 1559 | cytochrome P450 family 2 subfamily C member 9 | 9.154 | 1.209 | 1,20E-13 |
|  | *LDHAL6A* | 160287 | lactate dehydrogenase A like 6A | 9.043 | 1.214 | 3,20E-13 |
|  | *RSPH10B2* | 728194 | radial spoke head 10 homolog B2 | 8.982 | 1.213 | 4,30E-13 |
| down | *KRT17* | 3872 | Keratin 17 | -6.115 | 1.115 | 1,00E-07 |
|  | *RAB7B* | 338382 | member RAS oncogene family | -4.462 | 0,484 | 1,30E-19 |
|  | *IL6* | 3569 | Interleukin 6 | -3.943 | 0,23 | 1,40E-64 |
|  | *ARC* | 23237 | activity regulated cytoskeleton associated protein | -3.907 | 0,196 | 3,60E-87 |
|  | *CCNA1* | 8900 | cyclin A1 | -3.771 | 0,348 | 1,00E-26 |
|  | *SLPI* | 6590 | secretory leukocyte peptidase inhibitor | -3.742 | 0,622 | 4,90E-09 |
|  | *ST8SIA6* | 338596 | ST8 alpha-N-acetyl-neuraminide alpha-2,8-sialyltransferase 6 | -3.535 | 0,628 | 4,50E-08 |
|  | *C11orf86* | 254439 | chromosome 11 open reading frame 86 | -3.534 | 0,666 | 2,70E-07 |
|  | *MEST* | 4232 | mesoderm specific transcript | -3.417 | 0,584 | 1,30E-08 |
|  | *IL2RB* | 3560 | interleukin 2 receptor subunit beta | -3.387 | 0,319 | 1,40E-25 |

We performed gene set enrichment analysis (GSEA) with the co-expression gene set tmod and the Molecular Signatures Database (MSigDB) gene sets Reactome, Hallmark, KEGG, and GO, contrasting *CEBPB* knockdown cells to negative control transfected cells. The largest gene set of the gene set collections, which passed the significance threshold of $P_{adj} < 0.05$ and an area under the curve (AUC) > 0.7 indicated agonistic functions of *CEBPB* in the regulation of epithelial-mesenchymal transition and ECM interaction with AUC = 0.74 and $P_{adj} = 3.6 \times 10^{-40}$ (Table 2, Fig 5, **Appendix** Fig 3).

### *CEBPB* does not activate *SOST* expression in SaOS-2 cells

*CEBPB* knockdown did not reduce *SOST* expression (Fig 4C)**,** suggesting that *CEBPB* does not activate *SOST* expression. This is in contrast to our reporter gene experiment and GTEx data. As a next step, we asked whether the upregulation of *CEBPB* would affect the activation of *SOST* expression. The *CEBPB* gene encodes 2 functional isoforms (designated liver-enriched activating proteins LAP1 and LAP2) and the N-terminal truncated form liver-enriched inhibitory protein LIP, which lacks transactivation domains and is thought to inhibit the functions of LAP by competitively binding to the same DNA recognition sites [57]. To test whether LAP and LIP isoforms affect *SOST* expression, we overexpressed LAP2 and LIP in SaOS-2 cells for 48 h using episomal expression plasmids. LAP2 and LIP expression were upregulated 137- and 319-fold, respectively. Overexpression of the functional *CEBPB* isoform LAP2 showed a significant reduction of relative transcript levels, whereas overexpression of the truncated *CEBPB* isoform LIP did not induce changes in *SOST* expression (Fig 6A). This correlated with the results from *CEBPB* knockdown, which weakly induced *SOST* expression.

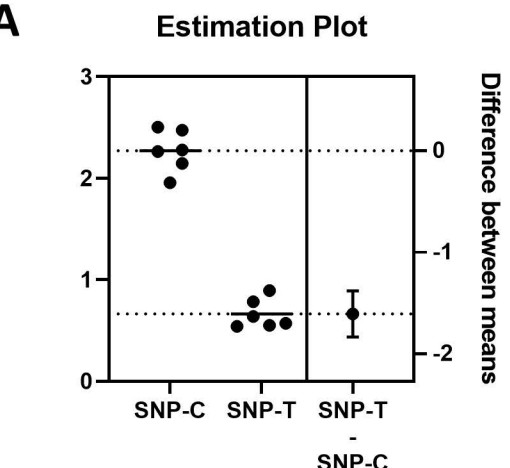
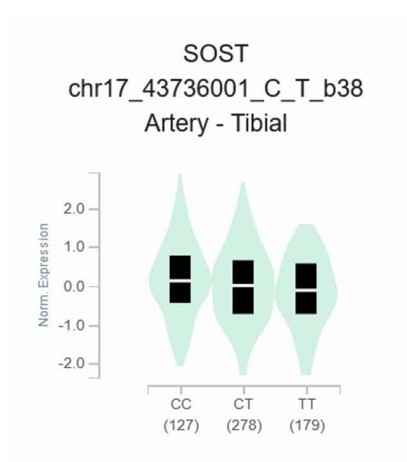

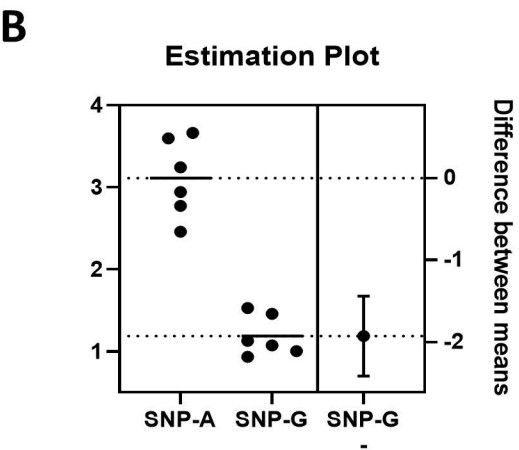
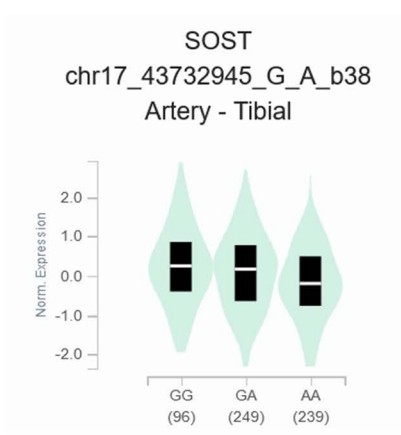

**Fig 3. rs9783823 and rs8071941 have allele-specific effects on luciferase activity.** The 79 bp sequence (39 bp up-and downstream of rs9783823) increased allele-specific enhancer activity in SaOS-2 cells in the background of the C-allele (2.3-fold stronger compared to the empty plasmid (baseline 1; P < 0.0001) and 3.8-fold stronger compared to the T-allele (P < 0.0001) (**A, left panel**). GTEx observed eQTL effects of rs9783823 on *SOST* expression (tissue: artery). Consistent with the reporter gene experiments, *SOST* expression was reduced in the background of the T-allele compared to the C-allele (normalized effect size β = -0.17, P = 5.1 x 10$^{-13}$; **A, right panel**). Both rs8071941 alleles increased luciferase activity (A: 3.0-fold, G: 1.3-fold, P = 0.0019; **B, left panel**). In contrast, GTEx data showed reduced *SOST* expression in the background of the A-allele (β = -0.24, P = 3.5 x 10$^{-25}$), consistent with the function of GFI-1 as a repressor (**B, right panel**). n = 3 biological replicates, each with a minimum of 2 technical replicates.

### *GFI-1* does not activate *SOST* expression in Hela cells

We tested whether *GFI-1* upregulation would affect *SOST* expression in Hela cells using CRIS-PRa. sgRNAs targeting the *GFI-1* promoter resulted in a 43-fold increase in *GFI-1* transcript levels. However, we observed no corresponding change in *SOST* transcript levels (Fig 6B).

## Discussion

In the current study, we identified the C-allele of rs9783823 as a biologically functional variant that impairs the binding of the TF CEBPB, with a putative causal contribution to

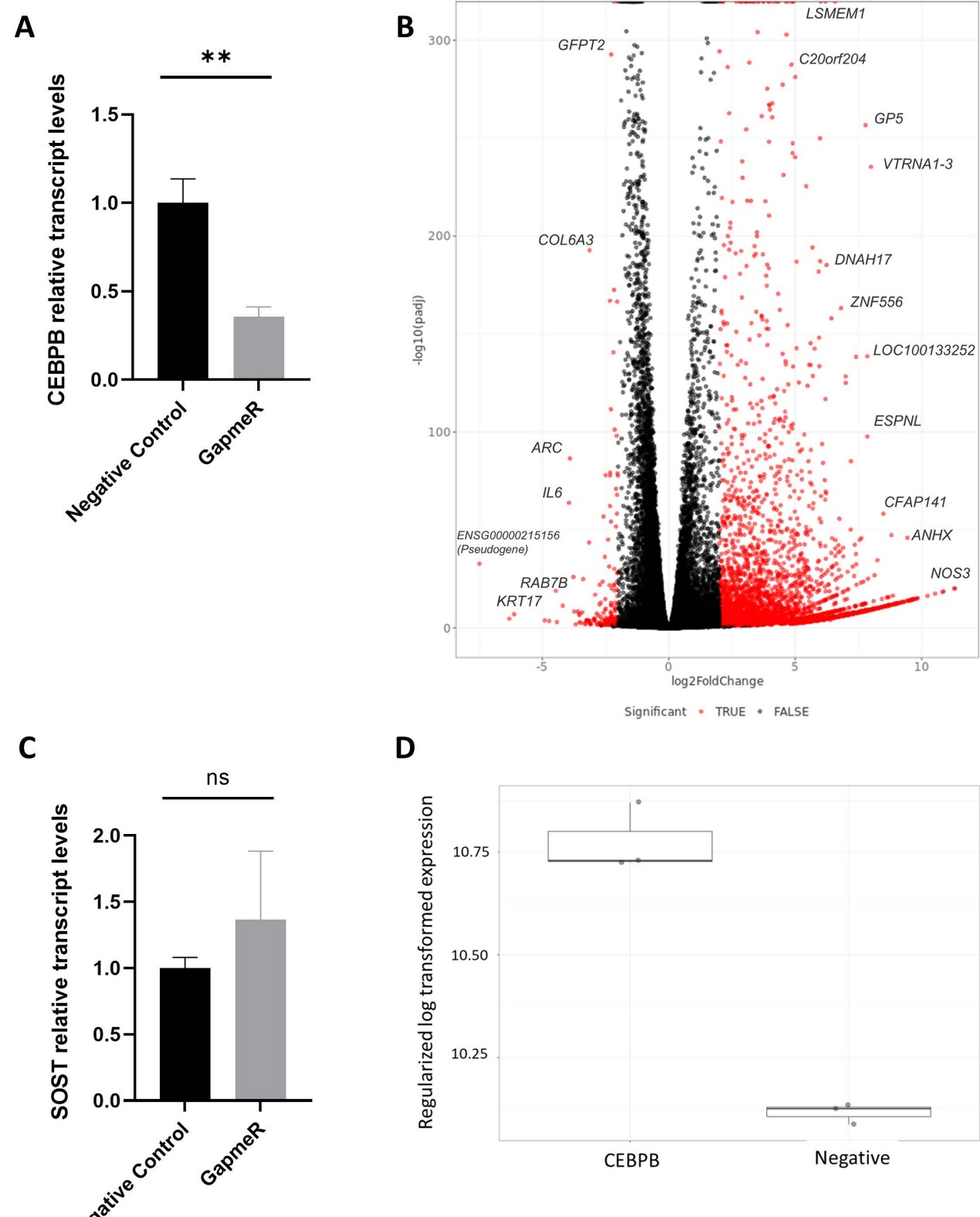

**Fig 4. *CEBPB* knockdown for 48 h primarily induced a significant upregulation of gene expression throughout the genome.** *CEBPB* expression was significantly reduced (0.36-fold, P < 0.01; **A**) 48 h after LNA GapmeR transfection. Volcano plot of SaOS-2 cells after *CEBPB* knockdown

primarily induced a significant upregulation of gene expression throughout the genome. Red color marks differentially expressed genes with P < 0.05 and Log2 FC > 2. The names of the most significant differentially expressed protein-coding genes are indicated (**B**). *CEBPB* knockdown in SaOS-2 cells resulted in a weak increase of *SOST* expression after 48 h. qRT-PCR with *CEBPB* specific primers (P = 0.29; **C**); RNA-Seq data (P = 4.7 x 10E-26, 0.79 log2FC; **D**). n = 3 experimental replicates.

**Table 2. Top Enriched Gene Sets after 48 h CEBPB knockdown in SaOS-2 cells.**

| Gene Set Collection | ID | Title | Gene Number | AUC | adj.P |
|---|---|---|---|---|---|
| Hallmark | M5930 | EPITHELIAL MESENCHYMAL TRANSITION | 190 | 0,74 | 3,60E-40 |
| KEGG | M7098 | ECM RECEPTOR INTERACTION | 71 | 0,73 | 1,60E-12 |
| Reactome | M27218 | NON INTEGRIN MEMBRANE ECM INTERACTIONS | 55 | 0,71 | 2,70E-09 |
| tmod | LI.M2.0 | ECM (I) | 30 | 0,77 | 5,30E-07 |
| GO | M12343 | REGULATION OF SUBSTRATE ADHESION DEPENDENT CELL SPREADING | 51 | 0,71 | 1,40E-06 |

increased alveolar bone loss of PIII-IV/C in smokers and reduced BMD. GTEx data showed that the rs9783823-C allele is significantly associated with increased *SOST* expression. Our data showed increased CEBPB binding in the background of the rs9783823-C allele as well as increased reporter gene expression, implying the functionality of this SNP. In addition, ChIP-Seq data from ENCODE confirmed CEBPB binding in this region in native chromatin and found 2 CEBPB ChIP-Seq peaks. We were able to show that IMR90 and A549 cells are homozygous for the T-allele of rs9783823, which explains the absence of a CEBPB ChIP-Seq peak at rs9783823. Remarkably, we found that the distance of the C-allele of the CEBPB motif at rs9783823 is identical to both ChIP-Seq peak 1 and peak 2, each with a distance of 1,427 bp. This suggests a possible stereometric function of the 3 CEBPB motifs. In general, only a small fraction of the sites for a given TF are bound by that TF, and this binding depends in part on the surrounding 3D environment [58], which may be reflected in the identical spacing of the three CEBPB binding sites. Taken together, these data suggest a role for CEBPB in the regulation of *SOST* expression. However, the upregulation of LAP2 in SaOS-2 cells did not alter *SOST* expression. This could be due to the strong baseline expression of *CEBPB* in SaOS-2 cells, which implies an a priori saturation of CEBPB binding sites. The ancestral allele of rs9783823 is C (C = 0.608 AFR) and the alternative allele is T. This implies a C → T transition. Those transitions occur spontaneously in the genome by oxidative deamination of 5-methylcytosine (5mC) to thymine (T). In most cases the resulting T:G mismatches are repaired. However, C:T mutations are enriched in the binding sites of CEBP, and it has been shown that within a CEBP site, the presence of a T:G mismatch increases the binding affinity of CEBP by a factor of >60 compared to the normal C:G base pair [59]. It has been suggested that this increased binding of CEBP to the T:G mismatch inhibits its repair. The passage of a replication fork over a T:G mismatch before repair can occur results in a C-to-T mutation in one of the daughter duplexes, providing a plausible mechanism for the accumulation of C-to-T somatic mutations in humans. In this context, the C and T alleles at rs9783823 indicate a CEBPB footprint of this TFBS, implying a biologically functional ancient CEBPB motif at this site. Our GSEA revealed that *CEBPB* knockdown repressed genes involved in epithelial-mesenchymal transition and ECM interaction. This suggests that CEBPB has an activating effect on these functions. This is consistent with previous studies that showed, for example, that knockdown of *CEBPB* in TNBC cells resulted in downregulation of genes involved in cell migration, extracellular matrix production, and cytoskeletal remodeling, many of which were epithelial-to-mesenchymal transition (EMT) marker genes [60]. In addition, CEBPB was

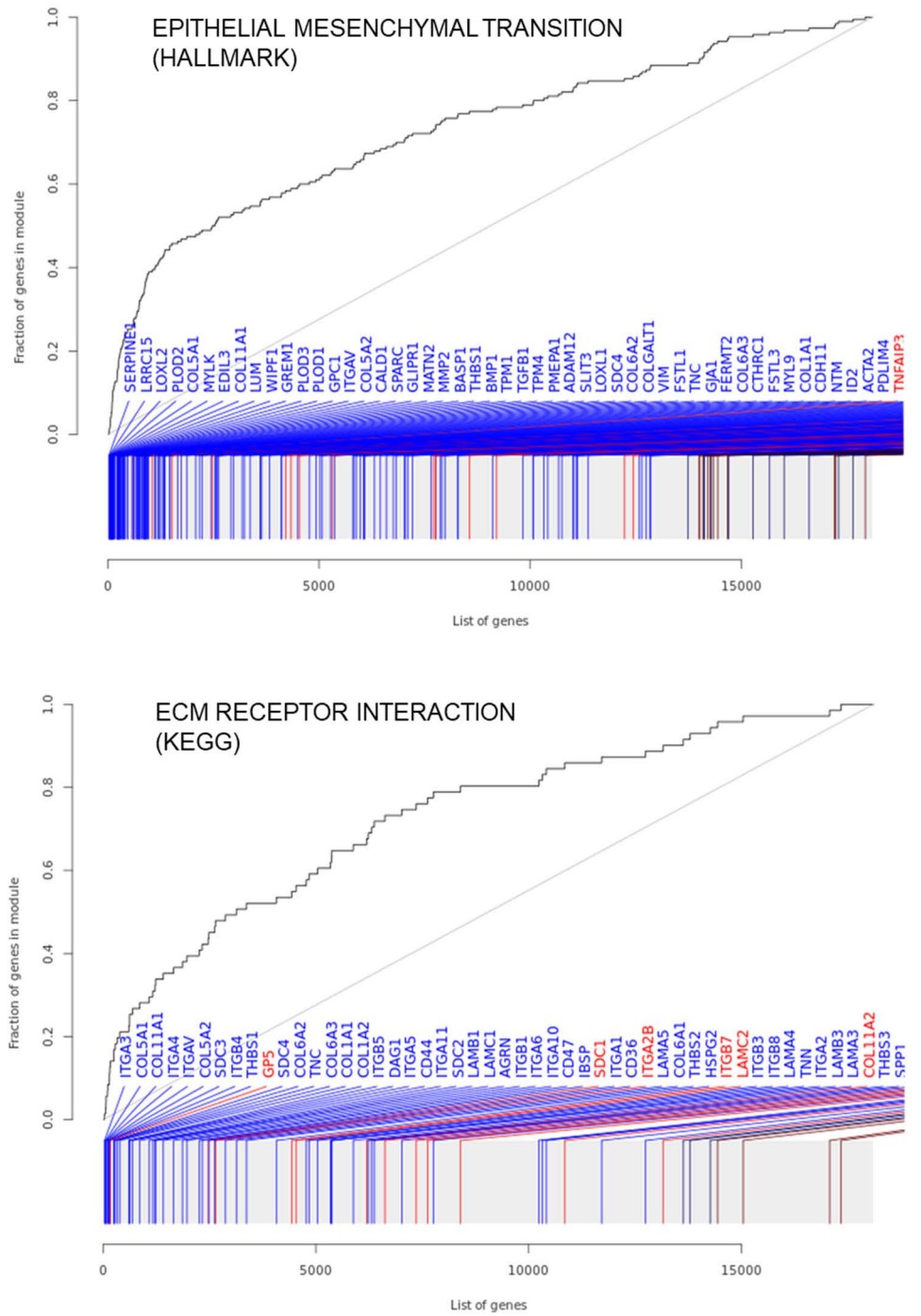

**Fig 5. Most significantly enriched gene sets after *CEBPB* knockdown in SaOS-2 cells for 48 h show a role in barrier tissue remodeling.**

identified as a master regulator of the mesenchymal phenotype, and overexpression of *CEBPB* in neural stem cells caused loss of neuronal differentiation, manifestation of a fibroblast-like morphology, induction of mesenchymal genes, and enhanced migration in a wound assay [61]. Conversely, siRNA- and shRNA-mediated *CEBPB* knockdown in SNB19 GBM cells

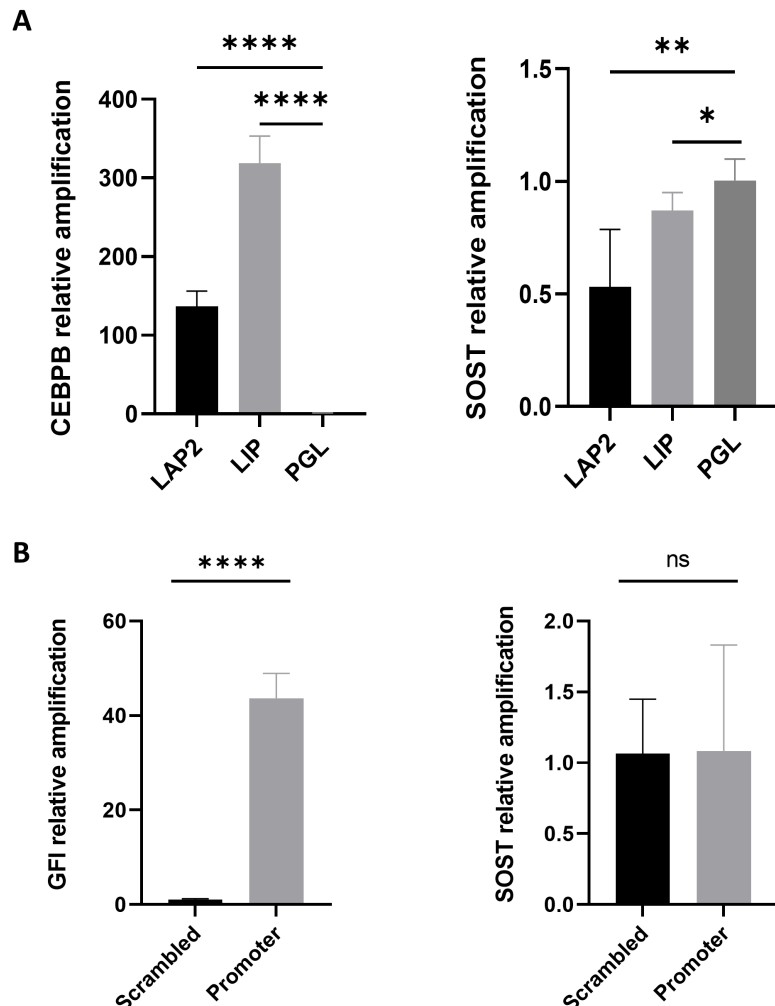

**Fig 6. Effect of overexpressing *CEBPB* and *GFI-1* on *SOST* expression. (A)** Overexpression of the *CEBPB*-LAP2 isoform reduced *SOST* expression in SaOS-2 cells. Significantly increased LAP2 and LIP expression 48 h after transfection of the corresponding overexpression plasmids (**left panel**). Increased LAP2 expression correlated with weakly reduced *SOST* expression (1.89-fold reduction, P < 0.01), and an increase in LIP expression showed a very slightly decreased *SOST* expression level (1.2-fold reduction, P < 0.05) (**right panel**). PGL = negative control vector. **(B).** *GFI-1* CRISPRa did not increase *SOST* expression in Hela cells. *GFI-1* was significantly upregulated 48 h after transfection of CRISPRa plasmids with sgRNAs that targeted the GFI promoter (43-fold, P < 0.0001; **left panel**). *SOST* expression did not show corresponding changes in transcript levels (P = 0.96; **right panel**). Each experiment was performed in biological triplicate.

showed suppression of the mesenchymal signature. At the single gene level, we observed that *CEBPB* knockdown correlated with a strong reduction of *KRT17* and *IL6* expression. This observation was also inconsistent with previous studies, which reciprocally showed that high *CEBPB* expression correlated with *KRT17* expression and the enhancer of this gene is regulated by CEBPB [62], and that overexpression of *CEBPB* resulted in a 10-12-fold increase in IL-6 production [63]. Our RNA sequencing data also showed that *CEBPB* knockdown was correlated with significantly reduced *SOST* expression. This supported our reporter gene data showing that CEBPB binding correlates with increased *SOST* expression. Notably, our RNA sequencing data showed that silencing of *CEBPB* correlated with ~ 6 times more upregulated genes than downregulated genes. However, our GSEA did not reveal any significant

enrichment of a gene set of upregulated genes. Instead, we found several downregulated gene sets of genes known to be regulated by *CEPBP*, such as epithelial-mesenchymal transition and ECM receptor interactions. We speculate that the detection of significantly downregulated gene sets as well as downregulated individual genes, some of which are already known targets of *CEBPB* regulation, indicate the true effects of *CEBPB* repression. In contrast, the comparatively higher number of genes with higher expression, paralleled by the lack of enriched gene sets after *CEBPB* repression, is likely due to indirect effects and possibly an associated perturbation of natural cell function. These non-physiological effects would explain the lack of enriched upregulated gene sets. In particular, we hypothesized that CEBPB would positively regulate *SOST* because our reporter gene experiments as well as the GTEx data implied higher gene expression in the background of rs9783823-C allele, which is the CEBPB binding allele. However, in our RNA sequencing data, we did not detect a downregulation of *SOST* after *CEBPB* knockdown, but a weak upregulation. Consistent with our interpretation that weakly upregulated genes in our RNA sequencing experiment reflect perturbation of natural cell function and are thus artifacts, we conclude that *CEBPB* does not positively regulate *SOST* expression in SaOS-2 cells. Therefore, we do not consider rs9783823 to be the causal variant of the association.

We also found a predicted conserved regulatory binding site of the transcriptional repressor GFI-1 at SNP rs8071941. It has previously been shown in knockout mice that loss of GFI-1 and housing under non-sterile conditions increases inflammatory response and bone mass loss [64]. In addition, GTEx showed that the GFI-1-binding rs8071941-A allele reduced *SOST* expression compared to the non-GFI-1-binding rs8071941-G allele (P = 5 x E-7, lung), consistent with the known repressor effect of GFI-1. However, we did not observe any effect of *GFI-1* overexpression on *SOST* expression after CRISPRa in SaOS-2 cells. Furthermore, GFI-1 binding to the genetic region of *SOST* has not yet been demonstrated by ChIP-Seq experiments from ENCODE and others [65,66]. Moreover, GFI-1 knockout mice showed reduced *SOST* expression [64], which is inconsistent with the effect of GFI-1 as a repressor of *SOST*. Taken together, we conclude that the causal variant of the haplotype block associated with periodontitis and BMD was not detected in this study.

A limitation of our study was that we did not examine all pleiotropic periodontitis- and BMD-associated SNPs, but limited the analyses to those SNPs that mapped to regulatory chromatin elements. Therefore, causal variants may be located in other regions of the LD block. However, TFBSs generally localize to active chromatin, as reliably detected by DHS, H3K27Ac, and H3K4Me1 histone modifications, as well as TFBSs experimentally confirmed by ChIP-seq. The results of our CRISPRa screen were consistent with this, showing the strongest gene expression in the promoter, followed by the nominated strong enhancer, while the nominated weak enhancer and chromatin not indicated as an active enhancer by these biochemical modifications had weak or no effect, respectively. Alternatively, the effect alleles do not necessarily have to alter TFBS to modulate gene transcription. However, we did not find SNPs in LD with periodontitis and BMD-associated haplotype block located in the 5' or 3' UTRs or at splice sites that could modulate gene transcription. In addition, we found no nonsynonymous SNP in an exon that would alter protein function. To identify the causal sequences underlying the association, deletion experiments of regions of the enhancer carrying the associated SNPs would be required.

In conclusion, we have identified an enhancer with strong cis-regulatory effects on *SOST* expression. This enhancer carries the CEBPB TFBS as determined by ChIP-Seq data from ENCODE. rs9783823 is flanked by 2 CHIP-Seq CEBPB binding sites and is likely to be a CEBPB binding site based on the spacing and G:T conversion. Regulation of *SOST* by CEBPB and GIF-1 has not been demonstrated.

## Supporting information

**S1 File. Title: SOST_Appendix.** Legend: This S1 File contains Appendix Material, Appendix Figures 1-3, and Appendix Tables 1-5 that provide additional data and analyses relevant to the main manuscript.
(DOCX)

## Acknowledgments

We thank Prof. Dr. Clemens Schmidt, Division of Cancer Genetics and Cellular Stress Responses, Max Delbrück Center for Molecular Medicine (MDC), Berlin, Germany, for providing cell pellets of IMR90 and A549 cells for sequencing.

## Author contributions

**Conceptualization:** Avneesh Chopra, Arne S. Schaefer.

**Data curation:** Avneesh Chopra, Jiahui Song, January Weiner 3rd, Dieter Beule, Arne S. Schaefer.

**Formal analysis:** Avneesh Chopra, Jiahui Song, January Weiner 3rd, Dieter Beule, Arne S. Schaefer.

**Funding acquisition:** Arne S. Schaefer.

**Investigation:** Avneesh Chopra, Jiahui Song, January Weiner 3rd, Dieter Beule, Arne S. Schaefer.

**Methodology:** Avneesh Chopra, Jiahui Song, January Weiner 3rd, Dieter Beule, Arne S. Schaefer.

**Project administration:** Avneesh Chopra, Arne S. Schaefer.

**Resources:** Arne S. Schaefer.

**Software:** Avneesh Chopra, Jiahui Song, January Weiner 3rd, Dieter Beule, Arne S. Schaefer.

**Supervision:** Avneesh Chopra, Arne S. Schaefer.

**Validation:** Avneesh Chopra, Arne S. Schaefer.

**Visualization:** Avneesh Chopra, Jiahui Song, January Weiner 3rd, Dieter Beule, Arne S. Schaefer.

**Writing – original draft:** Avneesh Chopra, Jiahui Song, Arne S. Schaefer.

**Writing – review & editing:** Avneesh Chopra, Jiahui Song, Arne S. Schaefer.

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
