## [Decision Letter · Decision Letter 0]

30 Jan 2025

Genetic Analysis of Cis-Enhancers Associated with Bone Mineral Density and Periodontitis in the Gene SOST

PONE-D-24-41083

PLOS ONE

Dear Dr. Chopra,

We’re pleased to inform you that your manuscript has been judged scientifically suitable for publication and will be formally accepted for publication once it meets all outstanding technical requirements.

Kind regards,

Md Shaifur Rahman, Ph.D

Academic Editor

PLOS ONE

Journal Requirements:

1. In your Methods section, please report the source of cell lines [ SaOS-2 cells and HeLa cells used for your study.

Reviewers' comments:

Reviewer's Responses to Questions

**Comments to the Author**

1. Is the manuscript technically sound, and do the data support the conclusions?

Reviewer #1: Yes

Reviewer #2: Yes

2. Has the statistical analysis been performed appropriately and rigorously? 

Reviewer #1: Yes

Reviewer #2: Yes

3. Have the authors made all data underlying the findings in their manuscript fully available?

Reviewer #1: Yes

Reviewer #2: Yes

4. Is the manuscript presented in an intelligible fashion and written in standard English?

Reviewer #1: Yes

Reviewer #2: Yes

5. Review Comments to the Author

Reviewer #1: The authors (Chopra et al) present recent work regarding the analysis of enhancer regions related to the SOST gene. This is an interesting area relating to the change of expression that could lead to diseases in human. Interestingly they identify enhancer regions that affect SOST expression. Their study goes on to look at the expression of other related genes in order to account for the mechanism and effects of this regulation. By doing this they present evidence of the involvement of specific transcription factors (C/EBPb). they further investigated the role of the transcription factor using a variety of techniques which added support to their original findings.

Abstract: well written, informative and relevant to the context of the manuscript.

Introduction: well written, informative and relevant to the investigation. As far as I can tell the references were appropriate

to the statements.

Materials and methods: comprehensive and informative without too much surplus information. The techniques used are standard within the field and are appropriate to the level of investigation. Experiments are performed to high technical standard.

Results: well written, informative and relevant.

Discussion: nicely presented and relevant to the results presented. The conclusions are interesting and I have no issue with proposed mechanism.

My only concern is the focus on the work was primarily done in cell lines which can in itself limit the validity of the conclusions, however it also included bioinformatic analysis of data available from databases albeit largely based on cell lines. It could be argued that the work would have benefitted from using primary cell lines to investigate expression using similar techniques to those described (eg dual-luciferase, silencing -qpcr), perhaps something like osteocytes/other relevant primary cells, or indeed animal models and remain within scope of the investigation. Other limitations were outlined in the discussion.

Very few obvious typos - limited to punctuation, parenthesis marks.

Meets criteria for publication and is recommended to be accepted.

Reviewer #2: Dear Author(s)

As an original research, the research meets the ethical standards, the experiments and analyses are appropriate for the focused question. The findings and data were also presented appropriately. The use of the language is also ok for publishing. So I have no other comments and suggestions through its acceptance and the manuscript can be published as its present version.

6. PLOS authors have the option to publish the peer review history of their article (what does this mean?). If published, this will include your full peer review and any attached files.

Reviewer #1: No

Reviewer #2: No

---

## [Editor Report · Acceptance letter]

PONE-D-24-41083

PLOS ONE

Dear Dr. Chopra,

I'm pleased to inform you that your manuscript has been deemed suitable for publication in PLOS ONE. Congratulations! Your manuscript is now being handed over to our production team.

Kind regards,

on behalf of

Dr. Md Shaifur Rahman

Academic Editor

PLOS ONE